

# Antibiotic susceptibility and resistance genes profiles of *Vagococcus salmoninarum* in a rainbow trout (*Oncorhyncus mykiss*, Walbaum) farm

Mesut Yilmaz[1], Tulin Arslan[2], Mükerrem Atalay Oral[3] and Aysegul Kubilay[4]

[1] Department of Aquaculture, Faculty of Fisheries, Akdeniz University, Antalya, Turkey
[2] Faculty of Fisheries, Mugla University, Muğla, Turkey
[3] Elmalı Vocational School of Higher Education, Akdeniz University, Antalya, Turkey
[4] Faculty of Eğirdir Fisheries, Isparta University of Applied Sciences, Isparta, Turkey

## ABSTRACT

Disease outbreaks negatively affect fish production. Antimicrobial agents used in the treatment of diseases become ineffective over time because of antibiotic resistance developed by bacteria distributed in the aquaculture environment. This study was conducted for 4 months (cold period) in a fish farm to detect the fish disease, cold water streptococcosis. In the study, four brood stock showing disease signs were detected. Bacteria isolates were obtained and identified as *Vagococcus salmoninarum*. Antimicrobial susceptibility of *V. salmoninarum* was tested and antibiotic resistance gene profiles of *V. salmoninarum* isolates were screened. The phylogenetic relation of the isolates with the previously reported strains was evaluated. Antibiotic resistance developed by pathogenic bacteria is distributed in the aquaculture environment. The transfer of resistance genes from one bacterium to another is very common. This situation causes the antimicrobial agents used in the treatment of diseases to become ineffective over time. The disc diffusion test showed that all four isolates developed resistance to 13 (FFC30, AX25, C30, E15, CF30, L2, OX1, S10, T30, CRO30, CC2, PT15 and TY15) of the evaluated antibiotics and were about to develop resistance to six others (AM 10, FM 300, CFP75, SXT25, APR15 and TE30). Furthermore, antibiotic resistance genes *tetA*, *sul1*, *sul2*, *sul3*, *dhfr1*, *ereB* and *floR* were detected in the isolated strain. Moreover, the phylogenetic analysis showed that isolated *V. salmoninarum* strain (ESN1) was closely related to the bacterial strains isolated from USA and Jura.

# INTRODUCTION

The importance of aquaculture in global nutrition is becoming emphasized. The global consumption of aquatic foods grew at about an average annual rate of 3.0 percent in the last six decades. According to a report by the Food and Agriculture Organization of the United Nations (*FAO, 2022*), global aquaculture production reached 122.6 million tones, of which 54.4 million tons was farmed in inland waters in 2020. One of the major finfish

Corresponding author
Mesut Yilmaz,
myilmaz@akdeniz.edu.tr

species is rainbow trout (*Oncorhynchus mykiss*) in inland aquaculture with 739,500 tones (1.5% of total production).

Disease outbreaks negatively affect fish production resulting in loss of billions of dollars (*Verschuere et al., 2000*; *Subasinghe, Bondad-Reantaso & McGladdery, 2001*). The development of efficient fish health management systems is essential for the success of an aquaculture operation (*Boran et al., 2013*). In recent years, management systems have been improved with the contribution of machine learning techniques (*Yilmaz et al., 2022*; *Yilmaz et al., 2023*; *Çakir et al., 2023*). Despite the advances in developing new vaccines, drugs and computer-assisted techniques, diseases are still the most devastating problem in fish farms (*Kusuda & Kawai, 1998*).

*Vagococcus salmoninarum* is a Gram-positive bacterium causing a chronic infection called 'cold water streptococcosis' (or vagococcosis) in rainbow trout broodstocks (*Michel et al., 1997*). The disease results in mass fish mortalities during the spawning period at water temperatures of 10–12 °C.

Vagococcosis was reported at fish farms in Australia (*Schmidtke & Carson, 1994*), the USA (*Standish et al., 2020*) and European countries such as Italy (*Ghittino et al., 2004*), France (*Michel et al., 1997*), Spain (*Ruiz-Zarzuela et al., 2005*; *Salogni et al., 2007*), Norway (*Schmidtke & Carson, 1994*) and Turkey (*Didinen et al., 2011*).

In Turkey, *V. salmoninarum* was first isolated from rainbow trout in the Mediterranean region in 2011 (*Didinen et al., 2011*). Then the disease was reported in the other regions of the country in 2012 (*Tanrıkul et al., 2014*) and in 2019–2020 (for five samplings) (*Saticioglu et al., 2021*). There are many studies on antibiotic resistance developed by bacteria distributed in the aquaculture environment (*Ruiz-Zarzuela et al., 2005*; *Michel et al., 1997*). The transfer of resistance genes from one bacterium to another (horizontal transfer) is very common (*Fu et al., 2022*). This situation causes the antimicrobial agents used in the treatment of diseases to become ineffective over time. On the other hand, fish farmers, fish processors and consumers may be exposed to various zoonotic infections due to contact with infected fish (*Gauthier, 2015*; *Sapkota et al., 2008*). While this risk group can be directly infected by pathogenic bacteria, resistance antibiotic genes can be transferable from fish pathogens to human pathogens by horizontal gene transfer (*Heuer et al., 2009*). Development of antibiotic resistance in pathogenic bacteria has become severe worldwide. Thus, it has taken its place among the priority areas of international organizations such as FAO.

In the present study, antimicrobial susceptibility, and antibiotic resistance gene profiles of *V. salmoninarum* isolates obtained from the diseased rainbow trout broodstocks were determined. The phylogenetic relation of the isolates with each other and the previously reported strains was evaluated.

## MATERIALS & METHODS
### Sample collection
During the monitoring study, broodstock (mean weight of 982.50 ± 176.85 g, $n = 4$) showing disease signs were collected from the concrete raceways of trout farms operating

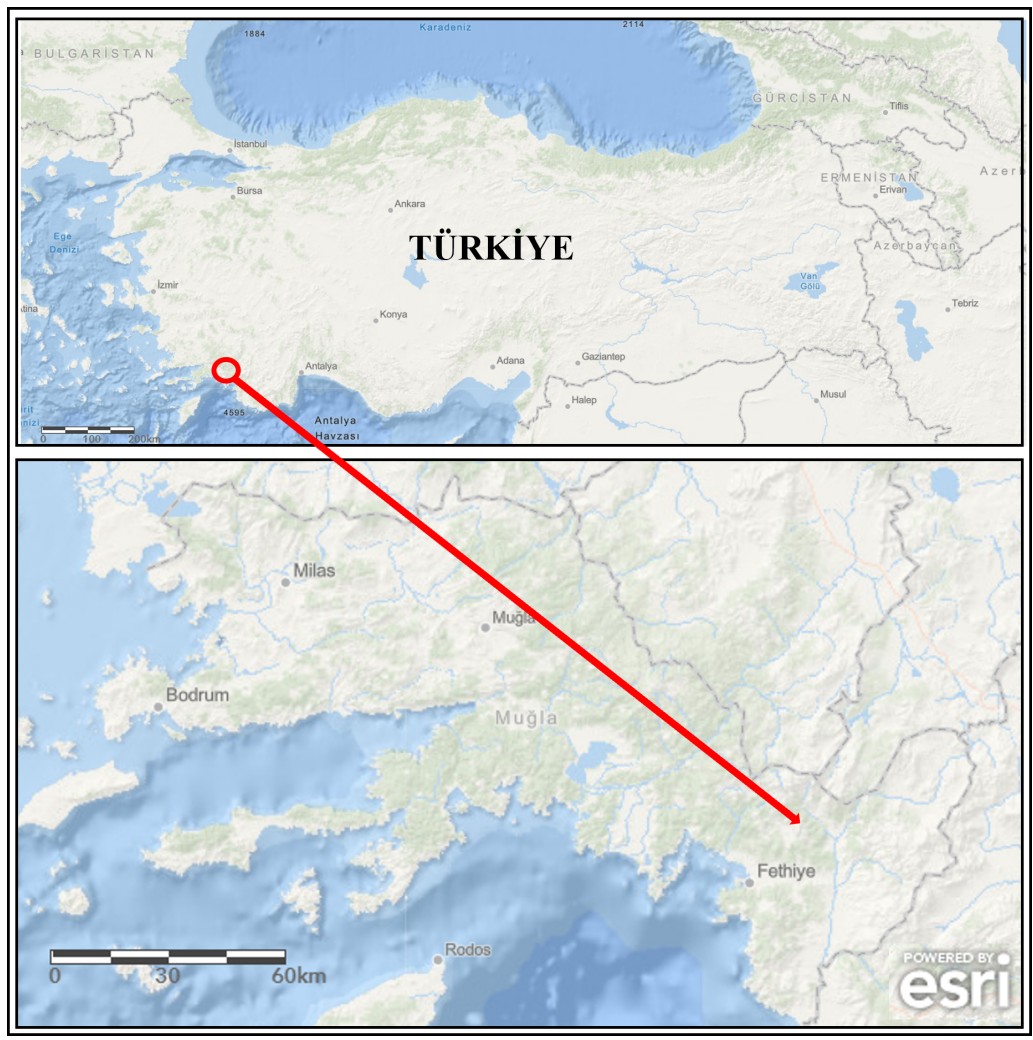

**Figure 1  The map of study area (the arrow is indicating location of the fish farm).** Powered by Esri, Redlands, CA, USA; https://www.arcgis.com.

in a large stream basin located in the southwestern part of Türkiye (Mugla province) (Fig. 1) from January to April (cold season) 2020. Specimens were collected with the permission of the Republic of Türkiye Ministry of Agriculture and Forestry, General Directorate of Agricultural Research and Policies (approval number: 92190712-605.99-E.5270). The diseased fish were first euthanized with MS-222 (100 mg/L) and then examined for external parasites and the symptoms (*Collins, 1993*). Afterwards, their skins were disinfected with 70% ethanol to perform necropsy. During necropsy, aseptic inoculates were taken from the liver onto plates containing Tryptic Soy Agar (TSA) medium. The inoculated TSA plates were incubated at $22 \pm 1$ °C for 48 h. Single-cell colonies were obtained from these inoculates by sub-culturing. Then, they were transferred into Tryptic Soy Broth (TSB) medium and incubated at $22 \pm 1$ °C for 24 h. The isolates grown in TSB were stored in 20% glycerol at $-80$ °C until used.

## Morphological and biochemical identification of the isolates

The single-cell colony samples stored in glycerol were planted on TSA plates and grown by incubating at $22 \pm 1$ °C for 24 h. After recording their characteristic features such as colony color, colony morphology, their biochemical characteristics were determined by Gram-staining, catalase, cytochrome oxidase, oxidation/fermentation (O/F) tests (*Austin & Austin, 2016*). A reference strain of *V. salmoninarum* (VSMed01/isolated strain in the study reported by *Didinen et al. (2011)* obtained from the Isparta University of Applied Sciences, Faculty of Eğirdir Fisheries (IFEF) Bacteria Collection was also included in the biochemical tests as a positive control.

## Molecular Identification of the Isolates

The single-cell colony samples stored in glycerol were cultured in 5 ml TSB medium overnight before DNA isolation. Approximately $1.5 \times 10^9$ cells (McFarland turbidity 0.5) were taken from these cultures as suggested in the DNA isolation kit (Thermo Fisher Scientific, Waltham, MA, USA) protocol. The quality and quantity of the isolated genomic DNA were evaluated by a nanodrop spectrophotometer (Thermo Fisher Scientific). The DNA isolates were stored at $-20$ °C until used.

Universal primers (B27F-5′AGAGTTTGATCCTGGCTCAG3′, U1492R-5′GGTTACCT TGTTACGACTT3′) were used for PCR of the targeted 16S rDNA sequence. The PCR reaction mix was prepared as recommended in the kit (Qiagen, Hilden, Germany) protocol. The PCR conditions for the amplification of 16S rDNA gene (about 1.4 kbp) was applied as an initial denaturation at 94 °C for 5 min, denaturation at 94 °C for 45 s, binding at 60 °C for 30 s, synthesis at 72 °C for 2 min and final elongation at 72 °C for 10 min. PCR reaction steps were repeated for 35 cycles for denaturation, annealing and synthesis steps. PCR products were run on the agarose gel (2%) to check product quality. During agarose gel electrophoresis, 8 volts/cm electric field was applied (*Brody & Kern, 2004*). Then, the PCR products were sequenced and compared to 16S rDNA sequences in the GenBank (*Chu & Lu, 2005*; *Liu et al., 2014*).

## Phylogenetic tree construction

The 16S gene sequence of *V. salmoninarum* isolated in this study and 13 different *V. salmoninarum* 16S gene sequences in the GenBank (X54272, MW622073, Y18097, MW151830, MK226201, KF012888, KF012889, JQ991578, MT452898, AM490375, MT452899 and AM490374) were used for the construction of phylogenetic tree. *Lactococcus garviae* (GenBank accession number: MW888490) was used as the out-group. Mr. Bayes (*Ronquist et al., 2012*) program was used for phylogenetic tree construction. According to the Bayesian Inference Criterion, the suitable mutation model for the sequences was determined as Hasegawa–Kishono–Yano (HKY) out of a total of 88 mutation models by using JmodelTest v.1.0. The analysis was carried out until the standard deviation of split frequencies fell under 0.01 (10,000,000 generations, sampling every 1,000) by setting The Markov chain Monte Carlo (MCMC) runs to four chains. The consensus trees were created using Figtree v. 1.4.2 after the first 25% of trees were removed as burn-in (*Rambaut, 2014*). Genetic distance analyses were conducted using the Kimura 2-parameter model (*Kimura,*

*1980*) by MEGA X (*Kumar et al., 2018*). All ambiguous positions were removed for each sequence pair (pairwise deletion option). The final dataset included a total of 702 positions.

## Antimicrobial susceptibility evaluation of the isolates

The single-cell colony samples stored in glycerol were planted into TSB medium and incubated at 22 ± 1 °C for 24 h. Following that, a physiological (0.9%) saline solution was used to bring the turbidity of the bacterial suspensions down to 0.5 McFarland turbidity (Biomerieux, Marcy-l'Étoile, France). A 100-μl adjusted sample was then spread out onto a Mueller-Hinton Agar (MHA) plate containing 5% sheep blood with a sterile cell spreader. Afterwards, the plates were dried for 5–10 min in a sterile cabinet and antibiotic discs were placed on the plates. For antibiotic sensitivity detection; ampicillin 10 μg (AM10), vancomycin 30 μg (VA30), trimethoprim/sulfamethoxazole 1.25/23.75 μg (SXT25), erythromycin 15 μg (E15), lincomycin 2 μg (L2), enrofloxacin 5 μg (ENO5), gentamicin 10 μg (GM10), tetracycline 30 μg (TE30), oxolinic acid 2 μg (OA2), ciprofloxacin 5 μg (CIP5), kanamycin 30 μg (K30), nitrofurantoin 300 μg (FM300), chloramphenicol 300 μg (C30), colistin 10 μg (CT10), doxycycline 30 μg (DOX30), ofloxacin 5 μg (OFX5), penicillin G 10 U (P10), streptomycin 10 μg (S10), clindamycin 2 μg (CC2), cefuroxime 30 μg (CXM30), oxytetracycline 30 μg (T30), cefoperazone 75 μg (CFP75), pristinamycin 15 μg (PT15), flumequine 30 μg (FLM30), tylosin 15 μg (TY15), ceftriaxone 30 μg (CRO30), spectinomycin 100 μg (SPT100), norfloxacin 10 μg (NOR10), cephalothin 30 μg (CF30), amoxicillin 25 μg (AX25), oxacillin 1 μg (OX1), apramycin 15 μg (APR15) and florphenicol 30 μg (FFC30) discs were used. Plates containing antibiotic discs were incubated for 24 h at 25 ± 1 °C. A digital caliper was used to measure the diameter of the inhibition zones that formed around the antibiotic discs at the end of the incubation period. Based on this measurement and related references (*Sezgin et al., 2023*), the isolates were categorized as susceptible, moderately susceptible, and resistant.

## Antibiotic resistance genes screening of the isolates

To determine antibiotic resistance at the genetic level, 18 antibiotic-resistance genes were screened by gene-specific primers (Table 1). The PCR reaction mixture was prepared as detailed in 'Phylogenetic Tree Construction'. The amplification of the targeted genes was carried out under the following PCR conditions: initial denaturation at 94 °C for 5 min, denaturation at 94 °C for 45 s, annealing at 55–68 °C (Table 1) for 30 s, synthesis at 72 °C for 1 min (90 s for *tetE*) and final elongation at 72 °C for 10 min. Denaturation, annealing, and synthesis steps were repeated for 35 cycles. The products were run on the agarose gel (2%) as described above to check the presence of the specific band belonging to the targeted resistance gene (*Brody & Kern, 2004*). Positive control bacterial strains (PCBS) (obtained from the IFEF Bacteria Collection) for antimicrobial resistance genes (Table 1) were used in all PCR analyses.

## Calculation of multiple antibiotic resistance index (MAR) value

The multiple antibiotic resistance (MAR) index value (*a/b*) was calculated as the ratio of the number of antibiotics to which the isolate was resistant (*a*) to the total number
**Table 1 Primer sequence, product size and AT information of the antibiotic resistance genes targeted PCR.**

| Targeted gene | PCBS collection number | Primer name | Primer nucleotide sequence (5′-3′) | EAS (bp) | AT (°C) | Reference |
|---|---|---|---|---|---|---|
| *tetA* | EAS4 | Tet A FW<br>Tet A RV | GCTACATCCTGCTTGCCTTC<br>CATAGATCGCCGTGAAGAGG | 210 | 60 | |
| *tetB* | ELG17 | Tet B FW<br>Tet B RV | TTGGTTAGGGGCAAGTTTTG<br>GTAATGGGCCAATAACACCG | 659 | 56 | |
| *tetC* | EAS4 | Tet C FW<br>Tet C RV | CTTGAGAGCCTTCAACCCAG<br>ATGGTCGTCATCTACCTGCC | 418 | 60 | *Ng et al. (2001)* |
| *tetD* | EAH13 | Tet D FW<br>Tet D RV | AAACCATTACGGCATTCTGC<br>GACCGGATACACCATCCATC | 787 | 56 | |
| *tet(E)* | ELG17 | Tet E FW<br>Tet E RV | GTGATGATGGCACTGGTCAT<br>CTCTGCTGTACATCGCTCTT | 1,180 | 58 | *Schmidt et al. (2001)* |
| *sul1* | EAH13 | Sul1 FW<br>Sul1 RV | CGGCGTGGGCTACCTGAACG<br>GCCGATCGCGTGAAGTTCCG | 433 | 65 | |
| *sul2* | ELG91 | Sul2 FW<br>Sul2 RV | GCGCTCAAGGCAGATGGCATT<br>GCGTTTGATACCGGCACCCGT | 293 | 65 | *Kern et al. (2002)* |
| *sul3* | ELG24 | Sul3 FW<br>Sul3 RV | TCAAAGCAAAATGATATGAGC<br>TTTCAAGGCATCTGATAAAGAC | 787 | 56 | *Heuer & Smalla (2007)* |
| *ampC* | EAS4 | AmpC FW<br>AmpC RV | TTCTATCAAMACTGGCARCC<br>CCYTTTTATGTACCCAYGA | 550 | 55 | *Schwartz et al. (2003)* |
| *blaTEM 1* | ELG91 | TEM OT-1 FW<br>TEM OT-2 RV | TTGGGTGCACGAGTGGGTTA<br>TAATTGTTGCCGGGAAGCTA | 465 | 58 | *Arlet & Philippon (1991)* |
| *blaSHV* | EAS4 | OS1<br>OS2 | TCGGGCCGCGTAGGCATGAT<br>AGCAGGGCGACAATCCCGCG | 862 | 68 | |
| *blaTEM 2* | ELG18 | TEM OT-3 FW<br>TEM OT-4 RV | ATGAGTATTCAACATTTCCG<br>CAATGCTTAATCAGTGAGG | 859 | 55 | *Olesen et al. (2004)* |
| *blaPSE* | ELG20 | PSE1 FW<br>PSE1 RV | CGCTTCCCGTTAACAAGTAC<br>CTGGTTCATTTCAGATAGCG | 465 | 58 | *Zühlsdorf & Wiedemann (1992)* |
| *dhfr1* | ELG91 | dhfr1-FW<br>dhfr1 RV | CTGATATTCCATGGAGTGCCA<br>CGTTGCTGCCACTTGTTAACC | 433 | 60 | *Schmidt et al. (2001)* |
| *aadA* | ELG91 | aadA FW<br>aadA RV | TGATTTGCTGGTTACGGTGAC<br>CGCTATGTTCTCTTGCTTTTG | 284 | 60 | *Van et al. (2008)* |
| *ereA* | ELG24 | ereA-F<br>ereA-R | AACACCCTGAACCCAAGGGACG<br>CTTCACATCCGGATTCGCTCGA | 420 | 65 | *Ounissi & Courvalin (1985)* |
| *ereB* | ELG91 | ereB-F<br>ereB-R | AGAAATGGAGGTTCATACTTACCACAT<br>ATAATCATCACCAATGGCA | 546 | 55 | *Arthur, Autissier & Courvalin (1986)* |
| *floR* | ELG17 | flor-F<br>flor-R | TATCTCCCTGTCGTTCCAG<br>AGAACTCGCCGATCAATG | 399 | 56 | *Van et al. (2008)* |

Notes.
  PCBS, positive control bacterial strain; EAS, expected amplicon size; AT, annealing temperature.

of antibiotics evaluated (*b*). Isolates with a determined MAR value greater than 0.2 were classified as having multiple antibiotic resistances (*Krumperman, 1983*).

## Ethical statement

The procedures applied in this study were evaluated by the Akdeniz University Animal Experiments Local Ethics Committee and their ethical compliance was approved with protocol number 737/2018.03.001.

## RESULTS

During the study, water quality parameters were evaluated by means of a multiparameter measuring device (YSI, ProPlus). They were all in good range for the rainbow trout reproduction (Temperature = 8.9 °C, pH = 8.24, Dissolved $O_2$ = 11.78 mg/L (saturation level of 98.70%), total dissolved solids = 0.15 g/L, salinity = 0.01%). Nevertheless, the farmers reported a 15% post-spawning mortality rate in sick broodstock. No parasite was detected on the external surface of 4 specimens ($n = 4$). Typical symptoms such as furuncles on the body surface, anemic gills, hyperemia and hemorrhage in visceral organs, splenomegaly were observed in all sick fish examined (Fig. 2). Furthermore, there were collapses and hemorrhages at the cranio-dorsal of the anal fin. The sick fish also had a hemorrhagic liver with nodules, an adherent and hemorrhagic intestine, a bloody and enlarged kidney, and bloody fluid accumulation in the intraperitoneal area.

## Morphological and biochemical identification of the isolates

A total of four inoculations were carried out from the livers of four brood fish sampled. All isolates obtained from four inoculations were Gram-positive, immobile, negative for catalase and cytochrome oxidase activity and fermentative in the O/F test (Table S1). Thus, all isolates were classified as *V. salmoninarum*.

## Molecular identification and phylogeny of the isolates

16S rDNA sequences of the four isolates were 100% compatible with each other. Thus, the 16S rDNA sequence was compared to the bacterial gene sequences in GenBank using the BLASTN tool. A perfect match (99.51%) was obtained with *V. salmoninarum* (GenBank accession number MW622073; maximal score 2,577, *E* value 0.0 and identity 1410/1417). The 16S rDNA sequence of the strain isolated in this study was submitted to GenBank with the accession number OK376503 as strain ESN1.

The phylogenetic tree generated over 16S rDNA sequences of 13 isolates showed that different *V. salmoninarum* strains were isolated from cultured rainbow trout worldwide. As indicated by the 100% bootstrap value, one strain from Argentina (MT452899) and one strain from Spain (AM490374) are well separated from the rest of *V. salmoninarum* strains, including the ESN1 strain defined in this study. These two distantly related *V. salmoninarum* strains settled together as another sister group (Fig. 3). The remaining 10 *V. salmoninarum* strains are clustered together with ESN1 with a very high (68%) bootstrap value. Additionally, genetic distance values ranged from 0.0014 to 0.0353 in the studied strains. While the lowest genetic distances were detected for strains from Turkey

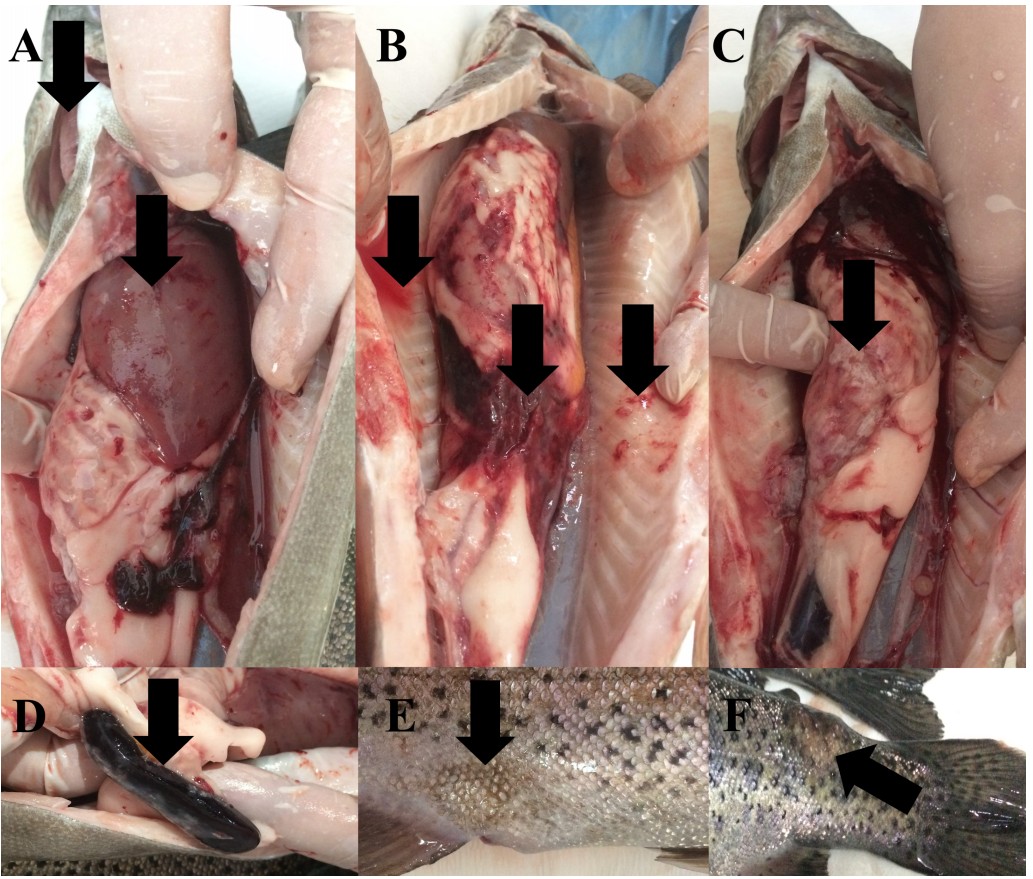

**Figure 2** **Infected broodstock.** (A) Anemic gills, hemorrhagic and anemic liver with nodules, (B) intraperitoneal bloody fluid accumulation, adherent and hemorrhagic intestine, and bleeding in the intra-abdominal capillaries, (C) damaged intestinal wall, (D) dark and enlarged spleen, (E) collapse at the craniodorsal of anal fin, (F) furuncle on the body.

(MW622073 and MW151830), USA (MK226201) and Jura (AM490375) with the value 0.0014, the highest genetic distance was observed for the Spanish strain (AM490374) with the value 0.0353. All three *V. salmoninarum* strains isolated from Turkey were closely related to the strains isolated from the USA and Jura compared to the strains isolated from other countries.

## Antibiotic resistance gene profiles of the isolates

Among 18 antibiotic resistance genes screened (Table 1), seven genes (*tetA*, *sul1*, *sul2*, *sul3*, *dhfr1*, *ereB* and *floR*) were detected in the ESN1 strain.

## Antimicrobial susceptibility of the isolates

The Kirby-Bauer disc diffusion test showed that all four isolates developed resistance to 13 (FFC30, AX25, C30, E15, CF30, L2, OX1, S10, T30, CRO30, CC2, PT15 and TY15) of the 33 antibiotics evaluated and were about to develop resistance to 6 others (AM 10, FM 300, CFP75, SXT25, APR15 and TE30). Yet, all four isolates were still susceptible to

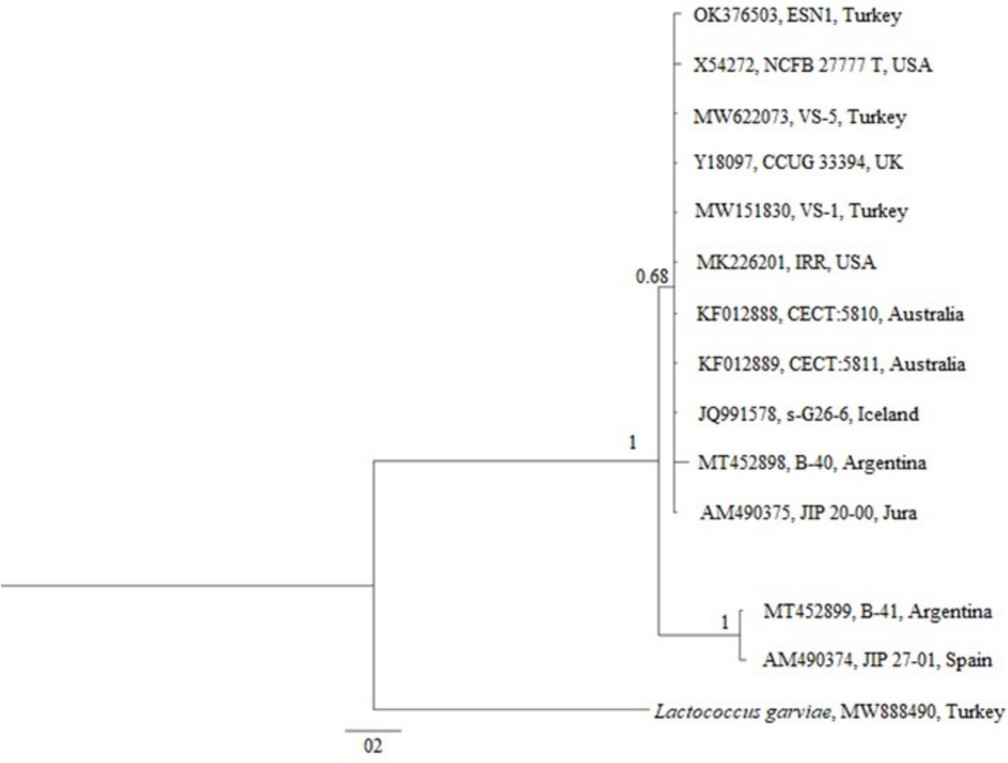

**Figure 3  Phylogenetic tree generated from 16S rDNA sequence.**

13 antibiotics (CIP5, GM10, CXM30, DOX30, ENO5, FLM30, K30, NOR10, OA2, OFX5, P10, SPT100 and VA30). We were not able to classify the susceptibility of isolates to one antibiotic (CT10) evaluated in this study due to unavailability of reference information, but we gave the inhibition zone diameter in Table 2.

MAR index was calculated as 0.4 which demonstrates that ESN1 has multiple antibiotic resistance.

## DISCUSSION

Of the 18 antibiotic resistance genes screened in this study, seven of them (*tetA*, *sul1*, *sul2*, *sul3*, *dhfr1*, *ereB* and *floR*) were present in the ESN1 strain. To the best of our knowledge, this is the second study evaluating the antibiotic-resistance gene profiles of a *V. salmoninarum* strain. The other study used a whole genome sequencing approach reported that VS-1 strain also isolated from Turkey carried 31 antibiotic resistance genes (*Saticioglu et al., 2021*). Nevertheless, only the *tetA* gene was shared between strains ESN1 and VS-1, and VS-1 did not have the other six antibiotic resistance genes detected in ESN1. The ESN1 strain may also have more than seven antibiotic resistance genes, but whole genome sequencing studies are also required in the ESN1 strain to see the full picture.

Concordantly, ESN1 strain showed resistance or about to acquire resistance to 57.6% of the 33 antibiotics tested in this study. Furthermore, ESN-1 was resistant to antibiotics

**Table 2** Antibiotic susceptibility of *Vagococcus salmoninarum* strains isolated from Turkey in different studies (Diameters of inhibition zones are given in parenthesis as mm).

| Antibiotics | *Didinen et al. (2011)* | *Tanrıkul et al. (2014)* | *Saticioglu et al. (2021)* | Present study |
|---|---|---|---|---|
| **AM10** | S | S | | I (16) |
| **AX25** | S | S | | R (12) |
| **OX1** | R | | | R (0) |
| **P10** | S | | | S (17) |
| **CF30** | | | | R (0) |
| **CFP75** | | | | I (19) |
| **CRO30** | | | | R (24) |
| **CXM30** | | | | S (18) |
| **SPT100** | | | | S (22) |
| **GM10** | | | R | S (24) |
| **K30** | S | | | S (20) |
| **S10** | R | | | R (0) |
| **SXT25** | | | R | I (14) |
| **VA30** | | | | S (20) |
| **CC2** | | | | R (0) |
| **L2** | S | | | R (0) |
| **CT10** | | | | - (19) |
| **APR15** | | | | I (20) |
| **E15** | S | R | | R (0) |
| **PT15** | | | | R (16) |
| **TY15** | | | | R (0) |
| **FM300** | | | | I (16) |
| **C30** | | | | R (11) |
| **FFC30** | S | S | | R (0) |
| **CIP5** | S | | R | S (36) |
| **ENO5** | S | S | R | S (36) |
| **FLM30** | | R | R | S (21) |
| **NOR10** | | S | | S (29) |
| **OA2** | | | R | S (27) |
| **OFX5** | | | | S (28) |
| **DOX30** | S | | | S (25) |
| **T30** | R | R | | R (0) |
| **TE30** | R | | | I (12) |
| **I+R/Total** | 4/14 | 3/8 | 6/6 | 19/33 |

**Notes.**

S, Susceptible; I, Moderately susceptible; R, Resistant.

-: unavailable reference information. The numbers in the abbreviated names show antibiotic concentration of the discs in μg, except penicillin G of which concentration was unit.

such as ampicillin, amoxicillin, lincomycin, florphenicol that previously isolated two *V. salmoninarum* strains from Turkey were found susceptible (*Tanrıkul et al., 2014*; *Didinen et al., 2011*). On contrary, it showed susceptibility to antibiotics such as gentamycin, ciprofloxacin, enrofloxacin, flumequine and oxolinic acid that VS-1 strain or another

previously isolated strain found resistant against them (*Saticioglu et al., 2021*; *Tanrıkul et al., 2014*).

As a result, it was stated that florphenicol can be used effectively in the treatment of vagococcosis. Also, it was experimentally demonstrated that florphenicol and doxycycline were more effective in treating vagococcosis infections in comparison to erythromycin and amoxicillin (*Kan & Didinen, 2016*). However, our disk diffusion test results showed that the ESN1 strain acquired resistance to 3 antibiotics, including florphenicol, but not to doxycycline.

The methodology employed in tests for antibiotic susceptibility can have an impact on the outcomes (*Sezgin et al., 2023*). In accordance with the most recent CLSI recommendations (*CLSI, 2021*), we employed 5% sheep blood-added MHA (SB-MHA) to assess the antibiotic susceptibility of the microorganisms.

In most of the reported studies, MHA were utilized for antibiotic susceptibility tests (*Didinen et al., 2011*; *Tanrıkul et al., 2014*). SB-MHA was employed in a recent study (*Saticioglu et al., 2021*). Sheep blood provides additional nutrients in the medium; therefore, SB-MHA promotes bacterial growth. Thus, SB-MHA can induce smaller inhibitory zones (*CLSI, 2021*), probably due to hemin in blood (*Nurjadi et al., 2021*). This demonstrates the need to follow a uniform approach to get comparable outcomes.

The outcome of our disc diffusion tests indicating that ESN1 strain is resistant or on the way of developing resistance to antibiotics such as oxytetracycline, tetracycline, trimethoprim/sulfamethoxazole, erythromycin, pristinamycin, tylosin, chloramphenicol and florphenicol was also supported by the presence of *tetA, dhfr1, ereB* and *floR* antibiotic resistance genes. On the other hand, we did not detect the antibiotic resistance genes in ESN1 such as *ampC, blaTEM 1, blaSHV, blaTEM 2, blaPSE, aadA*, which are thought to be responsible for detected resistance against antibiotics ampicillin, amoxicillin, oxacillin, cephalothin, cefoperazone, ceftriaxone and streptomycin (*Schwartz et al., 2003*; *Arlet & Philippon, 1991*; *Olesen, Hasman & Aarestrup, 2004*; *Zühlsdorf & Wiedemann, 1992*; *Van et al., 2008*). This implies the possibility that bacteria carry untested antibiotic resistance genes, or that acquired resistance may aid in the emergence of resistance to multiple antibiotics. Moreover, even the scanned *tetB, tetC, tetD, tetE, ereA* antibiotic resistance genes were not detected, *tetA* and *ereB* genes carried by the bacteria were possibly sufficient to provide resistance to related antibiotics (TE30, T30, E15, PT15 and TY15).

MAR was calculated as 0.4, which demonstrates that ESN1 has multiple antibiotic resistance. The results are compatible with the reported results (*Tanrıkul et al., 2014*; *Didinen et al., 2011*; *Saticioglu et al., 2021*; *Torres-Corral & Santos, 2019*). The isolate was also found to be resistant (13) or developing resistance (six) to over 50% of the 33 different evaluated antibiotics, most of which are commonly used in fish farms. The case indicates that the treatment of the disease can be quite difficult with these agents, as in a reported study (*Michel et al., 1997*).

The phylogenetic tree constructed by using 13 different 16S rDNA sequences deposited to the GenBank and that of the ESN1 strain described in this study showed that among all 13 of *V. salmoninarum* strains isolated from 8 different countries, including Turkey, the ESN1 was more closely related to a strains isolated from Turkey (MW622073 and MW151830),

USA (MK226201) and Jura (AM490375). Additionally, strains from Argentina and from Spain (MT452899 and AM490374, respectively) established a separate group, thus they were more distantly related to ESN1 and the remaining 10 strains. Similarly, *Saticioglu et al. (2021)* reported that *V. salmoninarum* isolates are divided into two clades.

The use of antibiotics used in the treatment of fish diseases by ignoring the primary factors causing the disease outbreak causes bacteria to become resistant to antibiotics over time (*FAO, 2006*). The use of antibiotic agents in this way causes the residues to disperse to different places, reach the environment and other living organisms, and thus spread antibiotic resistance. Antibiotic resistance in aquaculture areas has been demonstrated by various studies conducted in different years (*Miranda & Zemelman, 2002*; *Chelossi et al., 2003*; *Herwig, Gray & Weston, 1997*; *McPhearson et al., 1991*; *Schmidt et al., 2000*; *Twiddy, 1995*; *Aubry et al., 2000*; *Durán & Douglas, 2005*). For example, although an aquaculture farm did not use tetracycline in disease treatments, they found the *tetR* resistance gene in bacteria isolated from that farm (*Seyfried et al., 2010*). In addition, when isolates obtained from cultured and wild fish were compared in terms of antibiotic sensitivity, it was determined that isolates obtained from cultured fish were more resistant to antibiotics (*Akinbowale, Peng & Barton, 2006*; *Alcaide, Blasco & Esteve, 2005*). The findings obtained from our study are comparable with the reported studies mentioned above. The spread of antibiotic resistance among bacterial pathogens will make the use of existing antibiotics impossible over time. In such a situation, it becomes essential to develop new antibiotic agents, as it will not be possible to treat diseases with existing antibiotics. This vicious cycle clearly shows that unnecessary antibiotic use by ignoring the primary factors causing the disease outbreak in the farm environment threatens sustainable aquaculture.

## CONCLUSIONS

In conclusion, *V. salmoninarum* was isolated from the broodstock of a rainbow trout farm operating in the southwestern part of Turkey. We have limited knowledge of the *V. salmoninarum* relationship for epidemiological studies and determining optimal prophylactic measures against *V. salmoninarum* as an emerging fish pathogen. Vaccination is important for the prevention and control of these diseases.

Antibiotic resistance acquired by the pathogenic bacteria fails the antibiotic treatments and may cause important economic losses and unnecessary antibiotic use. For this reason, it is important to know the antibiotic resistance profile of a pathogenic bacterium before applying any antibiotic cure to treat sick fish. In addition, antibiotic resistance is easily acquired by bacteria and varies considerably between strains of a bacterium. For this reason, it would be more appropriate to perceive antibiotic resistance profiles, which are often reported as species-specific characters in the literature, as strain-specific characters. This ensures that the necessary legislative measures are taken to prevent the spread of identified antibiotic resistance genes. Our study has once again demonstrated the importance of monitoring fish farms for disease.

### Funding
The study was financially supported by the Scientific Research Projects Coordination Unit of Akdeniz University under the grant number of FBA-2018-3797. The funders had no role in study design, data collection and analysis, decision to publish, or preparation of the manuscript.

### Grant Disclosures
The following grant information was disclosed by the authors:
The Scientific Research Projects Coordination Unit of Akdeniz University: FBA-2018-3797.

### Competing Interests
The authors declare there are no competing interests.

### Author Contributions
- Mesut Yilmaz conceived and designed the experiments, performed the experiments, analyzed the data, prepared figures and/or tables, authored or reviewed drafts of the article, contributed reagents, materials, analysis tools, and approved the final draft.
- Tulin Arslan conceived and designed the experiments, performed the experiments, analyzed the data, prepared figures and/or tables, authored or reviewed drafts of the article, contributed reagents, materials, analysis tools, and approved the final draft.
- Mükerrem Atalay Oral conceived and designed the experiments, prepared figures and/or tables, authored or reviewed drafts of the article, and approved the final draft.
- Aysegul Kubilay conceived and designed the experiments, performed the experiments, prepared figures and/or tables, authored or reviewed drafts of the article, contributed reagents, analysis tools, and approved the final draft.

### Animal Ethics
The following information was supplied relating to ethical approvals (i.e., approving body and any reference numbers):
Akdeniz University Animal Experiments Local Ethics Committee and their ethical compliance was approved with the protocol number 737/2018.03.001.

### Field Study Permissions
The following information was supplied relating to field study approvals (i.e., approving body and any reference numbers):
The Ministry of Agriculture and Forestry is a government ministry of the Republic of Turkey and approved the study (92190712-605.99-E.5270).

### Data Availability
The data is available at NCBI: OK376503.

## Supplemental Information

Supplemental information for this article can be found online at http://dx.doi.org/10.7717/peerj.17194#supplemental-information.

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
