# Peer review of "Antibiotic susceptibility and resistance genes profiles of Vagococcus salmoninarum in a rainbow trout (Oncorhyncus mykiss, Walbaum) farm"

_PeerJ, doi:10.7717/peerj.17194_

## Round 0.1 · original submission · Major Revisions

The manuscript has been reviewed. The study is well presented and the methodology is correct. However, it contains numerous grammatical and spelling errors, so it is suggested that the English be thoroughly corrected throughout the text. Another very important consideration is that the figure legends are not correct. Please read the guidelines for authors and follow them in this and all sections of your manuscript.

Reviewers have detected other errors and suggested modifications. All of these should be considered in the revised manuscript.

**Language Note:** The Academic Editor has identified that the English language must be improved. PeerJ can provide language editing services - please contact us at [email protected] for pricing (be sure to provide your manuscript number and title). Alternatively, you should make your own arrangements to improve the language quality and provide details in your response letter. – PeerJ Staff

Reviewer 1 ·

Basic reporting

1. There are some English language corrections which need to be focused on:
Example:
a) Line 24: disease spelling
b) Line 117 & 169: run on the agarose gel.
c) Line 111: Primers were used for PCR of the targeted gene......

2. Reference missing- Line 44-45.

3. Some sentences need rephrasing- Line 67: The transfer of resistance genes from one bacterium to another (horizontal transfer) is very common.

4. Please include any reference of effects of consumption of such diseased trouts in human.
Try to briefly point out this aspect.

5. Line 249 & 260: resistant should be replaced by resistance.

Experimental design

1. Experimental design looks good.

2. Line 82: Please clarify, if it is January and April or January to April, since you mentioned that the study was conducted for a period of 4 months (Line 24).

Validity of the findings

1. Findings seems to be robust.

2. The mention of 'n' number would be helpful to determine the reproducibility of the data.

Additional comments

1. Line 261: I appreciate your awareness of how media might affect the resistance to certain drugs.
Sheep blood contains hemin, and there are references indicating how hemin plays a role in
determination of antimicrobial resistance.

Reviewer 2 ·

Basic reporting

Review for Yilmaz et al

This study focused on V. salmoninarum, an emerging pathogen in a rainbow trout farm's broodstock in southwestern Turkey. The objective was to assess the bacterium's antibiotic resistance and genetic profile, crucial for effective disease management in aquaculture.

Diseased broodstocks were collected, displaying symptoms like furuncles, anemic gills, and internal hemorrhages, but no external parasites. The bacteria isolated from these fish were identified as V. salmoninarum. The study's significant part involved examining the antibiotic resistance of the isolates. The found resistance to multiple antibiotics. of 18 screened antibiotic resistance genes, 7 (tetA, sul1, sul2, sul3, dhfr1, ereB, and floR) were present in the ESN1 strain.

Phylogenetic analysis showed that the V. salmoninarum strains, including ESN1, are related to global strains, indicating a wide geographical spread. The study highlights the importance of understanding antibiotic resistance at the strain level for managing diseases in aquaculture. It underlines the need for regular monitoring of fish farms and suggests the potential role of vaccinations in controlling infections. This research provides valuable insights into the genetic diversity and antibiotic resistance of V. salmoninarum, contributing to future disease control strategies in fish farming.

However, I have a major concern: the figure legends were not written appropriately, and the manuscript contained several grammatical errors."

Line 24: "diseae" should be spelled as "disease".

Lines 30-33: The abstract lists a large number of antibiotics and resistance genes without much context. For a general audience, it might be more informative to briefly explain the significance of these findings.

Line 35-36: The mention of "other countries (USA and Jura)" is somewhat vague. It might be clearer to specify the context or relevance of these geographical references.

Line 55: and onwards: The bacterium is referred to as "Vagococcus salmoninarum". Ensure that this is the correct and consistently used name throughout the study, as earlier sections mentioned "V. salmoninarum".

Line 200: The term "cytochrome oxidase" is misspelled as "cytochromoxidase"

Line 321: The term "realation" is incorrect; it should be "relation."

Line 322: What is profilactive? Is it prophylactic

Lines 322-323: Rephrase the sentence.
“…to epidemiologic study and to determine optimal profilactive prevention V. salmoninarum as emerging fish pathogen, vaccine is importance for prevention and control of the diseases."

to

"...for epidemiological studies and to determine optimal prophylactic measures against V. salmoninarum as an emerging fish pathogen. Vaccination is important for the prevention and control of these diseases."

Table 2 needs lot of improvement please aling all the antibiotics and susceptibility values.

Figure 2 Legends are missing.

Experimental design

no comment

Validity of the findings

no comment

Additional comments

This study focused on V. salmoninarum, an emerging pathogen in a rainbow trout farm's broodstock in southwestern Turkey. The objective was to assess the bacterium's antibiotic resistance and genetic profile, crucial for effective disease management in aquaculture.

Diseased broodstocks were collected, displaying symptoms like furuncles, anemic gills, and internal hemorrhages, but no external parasites. The bacteria isolated from these fish were identified as V. salmoninarum. The study's significant part involved examining the antibiotic resistance of the isolates. The found resistance to multiple antibiotics. of 18 screened antibiotic resistance genes, 7 (tetA, sul1, sul2, sul3, dhfr1, ereB, and floR) were present in the ESN1 strain.

Phylogenetic analysis showed that the V. salmoninarum strains, including ESN1, are related to global strains, indicating a wide geographical spread. The study highlights the importance of understanding antibiotic resistance at the strain level for managing diseases in aquaculture. It underlines the need for regular monitoring of fish farms and suggests the potential role of vaccinations in controlling infections. This research provides valuable insights into the genetic diversity and antibiotic resistance of V. salmoninarum, contributing to future disease control strategies in fish farming.

However, I have a major concern: the figure legends were not written appropriately, and the manuscript contained several grammatical errors."

---

## Round 0.2 · accepted · Accept

Many thanks for addressing all of the reviewers' comments. Now your manuscript is ready for publication in PeerJ.

Thank you for submitting your work to this journal.

Reviewer 1 ·

Basic reporting

1. All the concerns mentioned in my previous review have been addressed.

Experimental design

Nothing to add.

Validity of the findings

Nothing to add.

Additional comments

Nothing to add.

Reviewer 2 ·

Basic reporting

I appreciate the authors for their prompt and thorough revisions addressing my earlier concerns. The modifications have significantly strengthened the manuscript, and I am now satisfied with the completeness and clarity of the content.

Experimental design

'no comment'

Validity of the findings

'no comment'

Additional comments

'no comment'